# Dietary Iron Overload Differentially Modulates Chemically-Induced Liver Injury in Rats

**DOI:** 10.3390/nu12092784

**Published:** 2020-09-11

**Authors:** Mutsuki Mori, Takeshi Izawa, Yohei Inai, Sho Fujiwara, Ryo Aikawa, Mitsuru Kuwamura, Jyoji Yamate

**Affiliations:** Laboratory of Veterinary Pathology, Osaka Prefecture University, 1-58 Rinku Orai Kita, Osaka 598-8531, Japan; mu0411nya@gmail.com (M.M.); dc203013@edu.osakafu-u.ac.jp (Y.I.); sac01030@edu.osakafu-u.ac.jp (S.F.); szc01001@edu.osakafu-u.ac.jp (R.A.); kuwamura@vet.osakafu-u.ac.jp (M.K.); yamate@vet.osakafu-u.ac.jp (J.Y.)

**Keywords:** acute liver injury, apoptosis, ferroptosis, hepatic iron overload

## Abstract

Hepatic iron overload is well known as an important risk factor for progression of liver diseases; however, it is unknown whether it can alter the susceptibility to drug-induced hepatotoxicity. Here we investigate the pathological roles of iron overload in two single-dose models of chemically-induced liver injury. Rats were fed a high-iron (Fe) or standard diet (Cont) for four weeks and were then administered with allyl alcohol (AA) or carbon tetrachloride (CCl_4_). Twenty-four hours after administration mild mononuclear cell infiltration was seen in the periportal/portal area (Zone 1) in Cont-AA group, whereas extensive hepatocellular necrosis was seen in Fe-AA group. Centrilobular (Zone 3) hepatocellular necrosis was prominent in Cont-CCl_4_ group, which was attenuated in Fe-CCl_4_ group. Hepatic lipid peroxidation and hepatocellular DNA damage increased in Fe-AA group compared with Cont-AA group. Hepatic caspase-3 cleavage increased in Cont-CCl_4_ group, which was suppressed in Fe-CCl_4_ group. Our results showed that dietary iron overload exacerbates AA-induced Zone-1 liver injury via enhanced oxidative stress while it attenuates CCl_4_-induced Zone-3 liver injury, partly via the suppression of apoptosis pathway. This study suggested that susceptibility to drugs or chemical compounds can be differentially altered in iron-overloaded livers.

## 1. Introduction

The liver is the principal organ of drug metabolism, and it mediates activation or detoxification of various drugs and chemical compounds. Drug-induced liver injury (DILI) is one form of adverse drug reactions in the liver. Despite its low incidence in the general population, DILI is a frequent cause of acute liver failure and occasionally leads to sudden and life-threatening liver dysfunction [1]. The epidemiology of DILI varies by country: 13.9 new cases per 100,000 persons in France, 19 per 100,000 in Iceland, and 2.7 per 100,000 persons in the US [2]. DILI is caused by either predictable and dose-related mechanism such as direct toxicity of the drug or its metabolite, or unpredictable (idiosyncratic) mechanism. Multiple factors, such as sex, age, pre-existing liver diseases, or genetic factors, have been found to be associated with an increased susceptibility to idiosyncratic DILI [3]. However, the pathogenesis of idiosyncratic DILI remains unknown because of the complex interplay between drug and host factors. Thus, the research field of drug-induced hepatotoxicity has been expanding rapidly in order to identify mechanisms of hepatic injury and predict risks of pharmacologic agents.

Chronic liver disease (CLD) is a wide spectrum of diseases caused by various types of liver injury such as viral infection, alcoholic abuse, lifestyle-related diseases, and chemical toxicosis. At early stages, the liver can repair the damage/dysfunction; however, repeated and/or persistent injury triggers liver fibrosis, which progresses to cirrhosis and ultimately hepatocellular carcinoma (HCC) [4]. HCC is ranked as the sixth most common neoplasm and the third leading cause of cancer deaths in the world [5]. CLD due to hepatitis B or C accounts for the majority of HCC cases [6]. However, due to the increasing prevalence of metabolic syndrome, nonalcoholic fatty liver disease (NAFLD) and its progressive form nonalcoholic steatohepatitis (NASH) have recently been more common as the cause of CLD [7].

Iron is an essential micronutrient for many biological processes such as oxygen transport and storage, oxidative phosphorylation, and the catalysis of many metabolic redox reactions [8]. As mammals do not have major physiological pathway for iron excretion, iron uptake and storage are tightly regulated to avoid its excess [9]. The liver has an important role in systemic iron homeostasis. Hepatocytes produce hepcidin, a central regulator of systemic iron homeostasis. In the condition of CLDs, hepcidin production can decrease, resulting in secondary iron overload [10]. Increasing serum transferrin saturation to more than 75% can lead to an increase in toxic non-transferrin bound iron (NTBI) in the blood, which has a high potential for generating reactive oxygen species (ROS) [11]. Circulating NTBI is then taken up by hepatocytes. ROS increases cellular oxidative stress and leads to damage of the lipid membrane, protein, and DNA, triggering cell death such as necrosis or apoptosis. Thus, excessive accumulation of iron in the liver is an important risk factor for progression of CLDs [12]. However, it is unknown whether iron overload in CLD is a risk factor for DILI, in other words whether iron overload can alter the susceptibility to drug-induced hepatotoxicity.

Iron deposition in hepatocytes is proven to be more intense in the periportal area (Rappaport Zone 1) than in the centrilobular area (Zone 3), by the studies on human patients with hereditary hemochromatosis [13], and animals with experimental iron overload [14]. As hepatocytes have a zonal difference (zonation) in the abundance of metabolic enzymes (Zone 3 > 1), endoplasmic reticulum (Zone 3 > 1), glutathione (Zone 1 > 3), and mitochondria (Zone 1 > 3), the function of xenobiotic metabolism against hepatotoxicants can differ between Zones 1 and 3. In addition, cell types with iron deposition may vary between individual iron-overload diseases. For instance, accumulation of iron is marked in Kupffer cells in ferroportin disease [15], whereas it is prominent in hepatocytes in hereditary hemochromatosis [16]. Our previous study showed that dietary iron overload abrogates thioacetamide (TAA)-induced chronic liver cirrhosis in rats [14], raising the hypothesis that the susceptibility to chemically-induced liver injury is differentially modulated between Zone-1 and Zone-3 hepatocytes by hepatic iron overload.

Therefore, in order to elucidate the pathological roles of iron overload in chemically-induced liver injury, here we selected two single-dose models of acute liver injury: allyl alcohol (AA) and carbon tetrachloride (CCl_4_) models. Both chemicals are well-known hepatotoxicants inducing liver injury with intense oxidative stress. Importantly, the two hepatotoxicants differentially target zones of the liver lobule (Zone 1 in AA vs. Zone 3 in CCl_4_). The goal of the present study is to clarify the mechanism underlying the modulation of liver injury by comparing the data between the two different models.

## 2. Materials and Methods

### 2.1. Animals

Six weeks-old male F344/DuCrlCrlj rats (Charles River Laboratories Japan, Yokohama, Japan) were divided into control (Cont-saline), high-iron (Fe-saline), AA (Cont-AA), high-iron AA (Fe-AA), CCl_4_ (Cont-CCl_4_), and high-iron CCl_4_ (Fe-CCl_4_) groups as shown in Figure 1. Rats in Cont-saline, Cont-AA, and Cont-CCl_4_ groups were fed a regular diet (DC-8, containing 0.02% iron; CLEA Japan, Tokyo, Japan) while rats in Fe-saline, Fe-AA, and Fe-CCl_4_ groups were fed a high-iron diet (containing 0.8% iron; Oriental Yeast Co. Ltd., Tokyo, Japan) for 4 weeks. After the 4-week feeding, rats in Cont-AA and Fe-AA groups were intraperitoneally injected with AA (35 mg/kg) while rats in Cont-CCl_4_ and Fe-CCl_4_ groups were orally administrated with CCl_4_ (0.75 mL/kg). Rats in Cont-saline and Fe-saline groups intraperitoneally received an equivalent volume of physiological saline. At 24 h after the injection, rats were euthanized under deep isoflurane anesthesia, and the whole blood and liver were collected. Rats were maintained in a room with controlled temperature and 12-h light-dark cycle. Food and water were provided ad libitum. All experiments were approved by the Animal Care and Use Committee at Osaka Prefecture University (Osaka, Japan) (code nos. 27–184, 28–20, and 29–137) and were performed according to the Guidelines for Animal Experimentation of Osaka Prefecture University (Osaka, Japan).

### 2.2. Biochemical Analyses

Blood was collected from the abdominal aorta and serum was separated by centrifugation (3000 rpm, 10 min, 4 °C) as previously described [14]. Biochemical analyses were performed in SRL Inc. (Tokyo, Japan).

### 2.3. Histopathology

The left lateral and caudate lobe of the liver were fixed in 10% neutral-buffered formalin, embedded in paraffin, cut at 5 µm and stained with hematoxylin and eosin (HE) for histopathological examination. Iron histochemistry was performed with 3,3′-diaminobenzidine (DAB)-enhanced Perls’ stain as previously described [14].

### 2.4. Immunohistochemistry

Sections from the left lateral and caudate lobe of the liver were subjected to immunohistochemistry with primary antibodies specific for CD3, CD68, and phosphorylated-histone H2A.X (γH2A.X) as listed in Table 1. After dewaxing and pretreatment, tissue sections were immunostained in a Histostainer™ system (Nichirei Biosciences, Tokyo, Japan). Briefly, sections were treated with 5% skimmed milk in phosphate buffered saline (PBS) for 10 min, with each primary antibody at room temperature for 1 h, with 3% H_2_O_2_ in PBS for 15 min, and with horseradish peroxidase-conjugated secondary antibody (Histofine Simple Stain MAX PO^®^; Nichirei Biosciences, Tokyo, Japan) at room temperature for 30 min. Positive reactions were visualized with a DAB substrate kit (Nichirei Biosciences, Tokyo, Japan). Sections were counterstained lightly with hematoxylin. In order to evaluate hepatocellular apoptosis and necrosis, sections were subjected to terminal deoxynucleotidyl transferase dUTP nick end labeling (TUNEL) assay as previously described [14].

### 2.5. Cell Count

The number of immunopositive cells was counted. Five 20× fields of periportal/portal and centrilobular regions were separately evaluated from each animal. The data were presented as the number of positive cells per mm^2^.

### 2.6. RT-PCR

Real-time RT-PCR was performed to examine expression patterns of major inflammatory cytokines. Liver samples from the right medial lobe were immersed in RNA later regent (Qiagen, Hilden, Germany) for 2 days and stored at −80 °C before use. Total RNA was extracted using SV Total RNA Isolation System (Promega, WI, USA). Two-point-five microgram of total RNA was reverse-transcribed to cDNA by SuperScript VILO cDNA synthesis kit (Invitrogen, Carlsbad, CA, USA). Real-time PCR was performed with TaqMan gene expression assays (Life Technologies, Carlsbad, CA, USA) in a PikoReal Real-Time 96 PCR System (Thermo Scientific, Waltham, MA, USA) as previously described [17]. Details of probes are listed in Table 2. Eukaryotic 18sRNA was used as internal control. The data were analyzed with the 2^−ΔΔCT^ method.

### 2.7. Malondialdehyde (MDA) Assay

To examine lipid peroxidation change, hepatic MDA content was analyzed by thiobarbituric acid reactive substances (TBARS) method using an MDA Assay Kit (Northernwest Life Science Specialities, Vancouver, WA, USA) according to the manufacture’s instruction.

### 2.8. Glutathione-S-S-Glutathione/Glutathione-SH (GSSG/GSH) Quantification

To examine hepatic antioxidant activity, hepatic GSSG/GSH ratio was analyzed using a GSSG/GSH Quantification Kit (Dojindo, Kumamoto, Japan) according to the manufacturer’s instructions.

### 2.9. Western Blot

The liver samples from the right medial lobe were homogenized in a RIPA buffer (20 mM Tris-HCl pH 7.5, 150 mM NaCl, 1 mM EDTA, 1 mM EGTA, 1% NP-40, 0.1% deoxycholate, 0.1% SDS, 1 mM NaF, 0.1 mM Na_3_VO_4_, 1 mM PMSF, and proteinase inhibitor cocktail; Nacalai tesque, Kyoto, Japan). After centrifugation at 13,000× *g* for 10 min, the supernatant was mixed with an equal volume of 2× SDS sample buffer (125 mM Tris-HCl, pH 6.8, 4% SDS, 30% glycerol, and 10% 2-mercaptoethanol) and then boiled at 95 °C for 5 min. The protein concentration was determined by an absorption spectrometer using Bio-Rad Protein Assay (Bio-Rad Laboratories, Hercules, CA, USA). Samples were separated on 5–20% gradient polyacrylamide gels and transferred to polyvinylidene difluoride membranes (Bio-Rad Laboratories, Hercules, CA, USA). The membranes were incubated overnight at 4 °C with primary antibodies as listed in Table 3, followed by an incubation with peroxidase-conjugated secondary antibody (Histofine Simple Stain MAX PO^®^; Nichirei Biosciences, Tokyo, Japan) for 30 min. Signals were visualized with ECL prime (GE Healthcare, Little Chalfont, UK), and quantified with a luminescent image analyzer (LAS-4000; GE Healthcare, Little Chalfont, UK). The quantified data of cleaved caspase-3 were normalized by those of total caspase-3. The data of phosphorylated-receptor-interacting protein kinase 3 (p-RIP3) and glutathione peroxidase 4 (GPX4) were normalized by those of total protein bands stained with Coomassie brilliant blue (Appendix A) as expression of commonly-used loading controls such as β-actin, α-tubulin, and glyceraldehyde-3-phosphate dehydrogenase (GAPDH) changed between groups in our models.

### 2.10. Statistical Analysis

Data are presented as box-and-whisker plots with min to max range. Statistical analyses were performed using a Prism software, version 8 (GraphPad, San Diego, CA, USA) with Sidak’s multiple comparison. A value of *p* < 0.05 was considered statistically significant.

## 3. Results

### 3.1. Dietary Iron Overload Exacerbates AA-Induced Acute Liver Injury, While it Suppresses CCl_4_-Induced Acute Liver Injury

#### 3.1.1. Biochemical Findings

Serum iron increased in Fe-saline and Fe-AA groups compared with Cont-saline and Cont-AA groups, respectively (Figure 2A); there was no significant difference of serum iron between Cont-CCl_4_ and Fe-CCl_4_ groups (*p* = 0.12). Serum transferrin saturation and liver iron content increased by high-iron diet feeding in all administration groups (Figure 2B,C), indicating that hepatic iron overload occurs in all the three groups with high-iron diet feeding. The liver iron content did not differ significantly between the Fe-saline, Fe-AA and Fe-CCl_4_ groups. Serum transaminases (AST and ALT) increased in the AA model, while they decreased in the CCl_4_ model by dietary iron overload (Figure 2D,E). They also increased in Fe-AA and Cont-CCl_4_ groups compared with Fe-saline and Cont-saline groups, respectively. There was no increase in serum AST or ALT in Fe-saline groups, suggesting that the dietary iron overload in this model does not induce hepatocellular injury. In the liver of Fe-saline group, iron deposition was seen in both hepatocytes and sinusoidal nonparenchymal cells (Appendix A); the staining intensity within the single cell was more intense in sinusoidal cells than in hepatocytes while the distribution was more extensive in hepatocytes than in sinusoidal cells. Hepatocellular staining was more intense in the periportal (Zone 1) region than in the centrilobular (Zone 3) region (Figure 2F and Appendix A). Iron-positive sinusoidal cells were more abundant in Zones 1–2, consistent with the distribution of Kupffer cells.

#### 3.1.2. Pathological Findings

Grossly, the liver in Fe-saline group was brownish in color compared with that in Cont-saline group (Figure 3). The liver in Cont-AA group had no significant change, whereas the liver in Fe-AA group had extensive discoloration, suggestive of extensive necrosis. The liver in Cont-CCl_4_ and Fe-CCl_4_ groups was diffusely discolored with the hepatic lobules becoming more visible, with no significant differences between the two groups.

Histopathologically, no histopathological changes were seen in the liver of Cont-saline and Fe-saline groups (Figure 3), suggesting that the dietary iron overload alone does not induce necrosis, inflammation, or fibrosis in the liver. In Cont-AA group, mild infiltration of mononuclear cells was seen in the portal and periportal area (Zone 1). In Fe-AA group, extensive necrosis with hemorrhage and mononuclear cell infiltration were seen with the extent ranging from the periportal area (Zone 1) to multiple hepatic lobules. In Cont-CCl_4_ group, coagulative necrosis and vacuolar degeneration of hepatocytes with mild infiltration of mononuclear cells and neutrophils were seen in the centrilobular area (Zone 3); the hepatocellular necrosis and degeneration were less prominent in Fe-CCl_4_ group. In the Fe-CCl_4_ group, almost all hepatocytes within the centrilobular lesion had an intense iron accumulation, while a few hepatocytes within the centrilobular lesion had a mild to moderate iron accumulation in the Cont-CCl_4_ group (Appendix A).

We next analyzed inflammatory changes within the hepatic lesions by immunohistochemistry (Figure 4). A few CD3-positive T cells and CD68-positive macrophages/Kupffer cells were detected in the liver of Cont-saline and Fe-saline groups (Figure 4A–D). Compared with the saline administration groups, the number of CD68-positive macrophages/Kupffer cells increased in the periportal/portal region (Zone 1) of Cont-AA group, while the number of CD3-positive T cells and CD68-positive macrophages/Kupffer cells markedly increased in both the centrilobular and periportal/portal regions of Fe-AA group; the positive cell number was higher than that of Cont-AA group (Figure 4A–F,I,J). The number of CD3-positive T cells and CD68-positive macrophages/Kupffer cells increased in both the centrilobular and periportal/portal regions of Cont-CCl_4_ group, compared with that in Cont-saline groups (Figure 4A–D,G,K). The number of CD68-positive macrophages/Kupffer cells was lower in the centrilobular and periportal/portal regions of Fe-CCl_4_ group than in Cont-CCl_4_ group (Figure 4C,D,K,L). Irrespective of chemicals and iron overload, the number of CD68-positive macrophages/Kupffer cells was higher than that of CD3-positive T cells within the lesions, suggesting that macrophages/Kupffer cells are the major inflammatory cells infiltrating in the AA- and CCl_4_-induced acute liver injury.

We then analyzed hepatic expression of cytokine genes in terms of Th1/Th2 (T cells) and M1/M2 (macrophages) phenotypes, in order to characterize inflammatory response in our AA and CCl_4_ models (Figure 5). In the AA model, expression of Th1- or M1-related cytokines (TNFα, IL6, CCL2 and CXCL1) and Th2- or M2-related cytokines (IL4 and TGFβ1) increased significantly in Fe-AA group compared with that of Fe-saline group; the expression of TNFα, IL6, CCL2 and IL4 was higher in Fe-AA than in Cont-AA group. In the CCl_4_ model, expression of Th1- or M1-cytokines (TNFα, IFNγ, IL6, CCL2 and CXCL1) and Th2- or M2-cytokines (IL10 and IL4) increased in Cont-CCl_4_ group compared with that in Cont-saline group; the upregulation of TNFα, CXCL1 and IL10 was suppressed by iron overload (Fe-CCl_4_). Of the nine cytokine genes examined in this study, expression pattern of TNFα, IL6, CCL2 and IL4 was related with that of serum transaminases and inflammatory cells.

### 3.2. Enhanced Oxidative Stress is Involved in the Exacerbation of AA-Induced Liver Injury by Dietary Iron Overload, While Suppression of Apoptosis Can Be Involved in the Attenuation of CCl_4_-Induced Liver Injury

#### 3.2.1. Changes in Oxidative Stress

To evaluate oxidative stress conditions, we investigated hepatic content of MDA (a lipid peroxidation marker) and GSH (antioxidant marker). Hepatic MDA content significantly increased in Fe-AA group, compared with Cont-AA and Fe-saline groups (Figure 6A). Hepatic GSH contents did not change significantly between all groups (Figure 6B), while GSSG/GSH ratio was lower in Fe-CCl_4_ than Cont-CCl_4_ group (Figure 6C), suggesting that hepatic oxidative stress condition/response was to some extent different between the two groups. The GSSG/GSH ratio did not differ significantly between Cont-AA and Fe-AA groups despite the production of a large amount of MDA in the Fe-AA liver, suggesting that the liver in the Fe-AA group could not properly respond to the strong oxidative stress. We also performed immunohistochemistry for γH2A.X, a marker for DNA damage (double-strand break) [18]. The number of γH2A.X-positive hepatocytes significantly increased in the centrilobular and periportal regions of Fe-AA group compared with Cont-AA and Fe-saline groups (Figure 6D–G); positive hepatocytes were more frequently seen in the periportal area (Zone 1) than in the centrilobular area (Zone 3). The number of γH2A.X-positive hepatocytes in the centrilobular region (Zone 3) increased significantly in Fe-CCl_4_ but not in Cont-CCl_4_ group compared with Fe-saline and Cont-saline groups, respectively (Figure 6D,E,H,I); no significant difference was seen between Cont-CCl_4_ and Fe-CCl_4_ groups. Hepatic expression of heme oxygenase-1, a major antioxidant enzyme in the liver, increased significantly in Fe-AA group compared with that in Cont-AA and Fe-saline groups, and tended to increase in Fe-CCl_4_ group compared with that in Cont-CCl_4_ group (Appendix A).

#### 3.2.2. Changes in Cell Death

Finally, we investigated hepatic expression of cell death markers to understand the type (s) of cell death specifically involved in the exacerbation of AA-induced liver injury and the attenuation of CCl_4_-induced liver injury by iron overload. Hepatic expression of cleaved caspase-3, a central molecule for caspase-dependent apoptosis [19], significantly increased in Cont-CCl_4_ group compared with Cont-saline group, which was suppressed by the iron overload (Figure 7A,D). Phosphorylated-receptor-interacting protein 3 (p-RIP3; a marker for necroptosis [20]), did not change significantly between all groups (Figure 7B,D). Expression of glutathione peroxidase (GPX4), a key regulator of ferroptosis [21], significantly decreased in Cont-AA and Fe-AA groups compared with Cont-saline and Fe-saline groups, respectively (Figure 7C,D); it also significantly decreased in Fe-CCl_4_ group compared with Fe-saline group. In order to investigate the types of cell death involved in the AA- and CCl_4_-induced liver injury, TUNEL assay was performed. The TUNEL method can label apoptotic hepatocytes with solely nuclear staining (reflecting nuclear DNA fragmentation) and necrotic hepatocytes with both nuclear and diffuse cytoplasmic staining, reflecting cytoplasmic transition of DNA fragments [22]. TUNEL-positive hepatocytes were few in Cont-AA group (Figure 7E) while most of hepatocytes in the periportal (Zone 1) lesion had an intense nuclear and cytoplasmic staining (Figure 7F), suggesting that necrotic cell death is mainly involved in the AA-induced liver injury with dietary iron overload. In the Cont-CCl_4_ group, apoptotic hepatocytes with solely nuclear staining and necrotic hepatocytes with both nuclear and cytoplasmic staining were scattered in the centrilobular (Zone 3) lesion (Figure 7G), suggesting that both apoptotic and necrotic cell death can be involved in the CCl_4_-induced liver injury. The number of apoptotic and necrotic hepatocytes tended to decrease in Fe-CCl_4_ group (Figure 7H) compared with that in Cont-CCl_4_ group (Figure 7G).

## 4. Discussion

In this study, we showed that dietary iron overload exacerbates AA-induced acute liver injury, characterized by increased serum transaminases and extensive necrosis with Zone 1 dominant inflammatory responses. Additionally, iron overload significantly increases hepatic lipid peroxidation and hepatocellular DNA double-strand break, associated with a decreased hepatic expression of anti-ferroptosis enzyme. These data suggest that enhanced oxidative stress is involved in the exacerbation of AA-induced liver injury by the iron overload. We also showed that dietary iron overload attenuates CCl_4_-induced liver injury, characterized by decreased serum transaminases, decreased centrilobular (Zone 3) necrosis and apoptosis, and weakened inflammatory responses. Iron overload abrogates caspase-3 cleavage induced by CCl_4_ administration. Despite the attenuation of liver injury, iron overload increases DNA double-strand break of centrilobular (Zone 3) hepatocytes with a mild increase in hepatic lipid peroxidation. These data suggest that suppression of caspase-dependent apoptosis can be involved in the attenuation of CCl_4_-induced hepatotoxicity by iron overload.

The pattern of hepatic iron deposition varies between individual iron overload diseases. In ferroportin disease, the iron accumulates mainly in the reticuloendothelial system (RES) cells [15], while it accumulates in the liver parenchyma in hereditary hemochromatosis [16]. Hepatic iron deposition secondarily occurring in chronic liver diseases can be divided into three patterns: a hepatocellular pattern, an RES pattern, or a mixed pattern involving both hepatocytes and RES cells [23]. Iron deposition in patients with NAFLD may occur with any of the three patterns, with the mixed pattern being most common [23]. In our dietary iron overload model, iron deposition was seen in both RES cells and hepatocytes, which resembles a mixed pattern of iron deposition in patients with CLD. Thus, we consider that our model would be useful to examine how iron deposition can affect susceptibility to drug/chemical-induced hepatotoxicity in the condition of secondary iron overload. Further comparative study using animal models with different patterns of hepatic iron deposition would clarify cell-specific roles in the modulation of hepatoxicity.

AA is protoxicant and metabolized by alcoholic enzymes to acrolein, a highly reactive molecule with two reactive centers consisting of a carbon-carbon double bond and an aldehydic group [24]. Acrolein induces extensive lipid peroxidation and thereby induces cell death by intense oxidative stress. Excess iron also can induce oxidative stress via generation of ROS mainly through the Fenton reaction, which eventually forms hydroxyl radicals from superoxide or hydrogen peroxide [25]. A candidate mechanism underlying the exacerbation of AA-induced liver injury in iron-overloaded liver is a combination of increased oxidative stress and increased ferroptosis. Ferroptosis is a recently identified form of regulated cell death mediated by iron-dependent ROS generation. It is morphologically, biochemically, and genetically different from other forms of cell death including apoptosis, necrosis, and autophagy [26]. Ferroptosis is triggered by inactivation of cellular GSH-dependent antioxidant defenses, leading to the accumulation of ROS and lipid peroxidation [27]. In our AA model, administration of AA alone induced a mild hepatic inflammation with a lack of histological necrosis and oxidative stress induction, while AA administration combined with dietary iron overload induced severe liver injury with a significantly enhanced lipid peroxidation and hepatocellular DNA damage. Interestingly, hepatic expression of GPX4, a central molecule for inhibition of ferroptosis-related lipid peroxidation, was decreased both in Cont-AA and Fe-AA groups, despite the lack of substantial change in lipid peroxidation in Cont-AA group. These finding raises the possibility that AA administration at the present dose (35 mg/kg) increases the susceptibility to ferroptosis and thus combination of AA administration with dietary iron overload (the major ferroptosis inducer) triggers ferroptosis-related extensive liver injury.

CCl_4_ is a hepatotoxicant and converted to trichloromethyl and trichloromethyl peroxy radicals by P450 enzymes such as CYP2E1 [28]. These radicals bind to antioxidant enzymes, including the sulfhydryl groups of GSH [29]. Overproduced free radicals increase membrane lipid peroxidation, bind covalently to macromolecules, deplete cellular ATP, and interfere with calcium homeostasis. As described above, liver injury was attenuated in Fe-CCl_4_ group despite the increased Zone-3 hepatocellular DNA damage; however, the degree of increased lipid peroxidation was modest in Fe-CCl_4_ group compared with Fe-AA group. In addition, hepatic GSSG/GSH ratio, an indicator for oxidative stress, was lower in Fe-CCl_4_ group than in Cont-CCl_4_ group, suggesting a suppression of oxidative stress in Fe-CCl_4_ group. It is known that moderate amount of oxidative stress can activate anti-oxidant response processes such as nuclear factor-erythroid 2-related factor 2 (Nrf2) [30]. Nrf2 is a transcription factor which induces transcription of antioxidant response elements mediated transcriptional activation, resulting in the production of numerous antioxidant enzymes, such as glutathione-s-transferase, glutathione peroxidase, superoxide dismutase, catalase, and heme oxygenase-1 (HO-1). In the present study, hepatic expression of HO-1 tended to increase in Fe-CCl_4_ group compared with Cont-CCl_4_ group; however, the increase was less prominent than in Fe-AA group, suggesting that other antioxidant enzymes or other antioxidant response could be involved in the attenuation of oxidative stress in the CCl_4_ model.

The suppression of caspase-dependent apoptosis is considered to contribute in part to the attenuation of CCl_4_-induced liver injury in this study. Our previous study also showed that suppression of apoptosis can contribute to abrogation of TAA-induced chronic liver cirrhosis in iron-overloaded rats [14]. A few studies have reported that iron can prevent apoptosis induced by multiple stresses [31,32,33]. As iron is essential for the function of mitochondrial function such as respiratory chain [34], cellular iron accumulation could have an impact on the balance between mitochondrial anti-apoptotic and pro-apoptotic molecules, such as B-cell lymphoma-2 and Bcl-2-associated X protein. Further mechanistic study is required to understand the detailed mechanisms of this unique phenomenon.

GSH is well-known as a potent antioxidant that interacts with ROS; GSH depletion is an important process in AA-induced liver injury [35]. Additionally, GSH is an essential co-factor for GPX4-mediated inhibition of ferroptosis [36]. However, hepatic GSH content did not change significantly by AA or CCl_4_ administration or iron overload in this study. Therefore, GSH is considered to be less involved in the modification of CCl_4_ or AA-induced liver injury in this model.

Both AA and CCl_4_ can induce hepatic injury by intense oxidative stress such as lipid peroxidation. However, AA-induced Zone-1 liver injury is exacerbated by dietary iron overload, while CCl_4_-induced Zone-3 liver injury is suppressed. Thus, there should be certain zone-specific mechanism underlying the modulation of chemically-induced liver injury by iron overload. Iron deposition in hepatocytes is more intense in the periportal area (Zone 1), the target region of AA-induced hepatotoxicity, than in the centrilobular area (Zone 3) in dietary iron overload rats [14]. As iron is localized preferentially in the cytoplasm of periportal (Zone 1) hepatocytes in the human newborn liver, Zone-1 hepatocytes may be mainly involved in iron storage in the physiological state [37]. It is also reported that gradient transferrin receptor distribution from Zones 1 to 3 hepatocytes causes the zonal difference in the degree of iron deposition in iron overload rats [38]. Therefore, it is likely that more intense iron deposition can enhance iron-mediated oxidative stress in Zone 1, while less intense iron deposition can protect Zone-3 hepatocytes against some chemically-induced hepatotoxicity. In addition to the iron distribution, oxygen partial pressure is higher in the periportal area (Zone 1) than in the centrilobular area (Zone 3), and the iron-catalyzed ROS are unavoidable in an oxygen-rich environment [39]. Laser microdissection-based zonal transcriptomic analysis would be helpful to further understand the molecular mechanisms of this difference in the modulation of chemically-induced liver injury between Zones 1 and 3.

## 5. Conclusions

This study showed that dietary iron overload exacerbates AA-induced Zone-1 acute liver injury and that enhanced oxidative stress-induced cell death, presumably via ferroptosis-related lipid ROS, can be responsible for the exacerbation. It also showed that dietary iron overload attenuates CCl_4_-induced Zone-3 acute liver injury, at least partly via suppression of apoptosis pathway. These data suggest that susceptibility to drugs or chemical compounds can be altered in iron-overloaded livers. As the majority of studies are focusing on exacerbation of liver diseases by iron overload, little is known about the mechanism underlying attenuation of liver diseases by iron overload. Further study on the zone-specific molecular characterization would give new insights into the pathophysiological role of iron overload in liver diseases.

## Figures and Tables

**Figure 1 nutrients-12-02784-f001:**
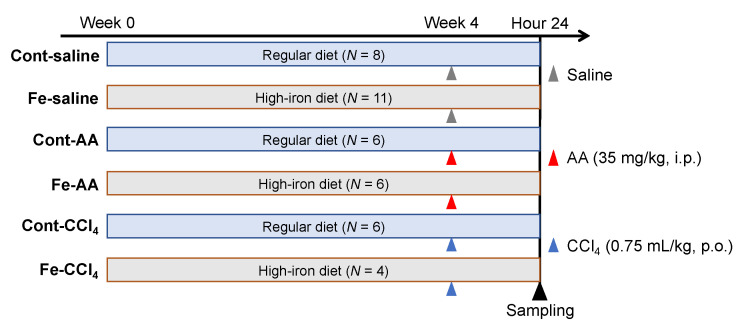
Experimental design of this study. AA; allyl alcohol, CCl_4_; carbon tetrachloride.

**Figure 2 nutrients-12-02784-f002:**
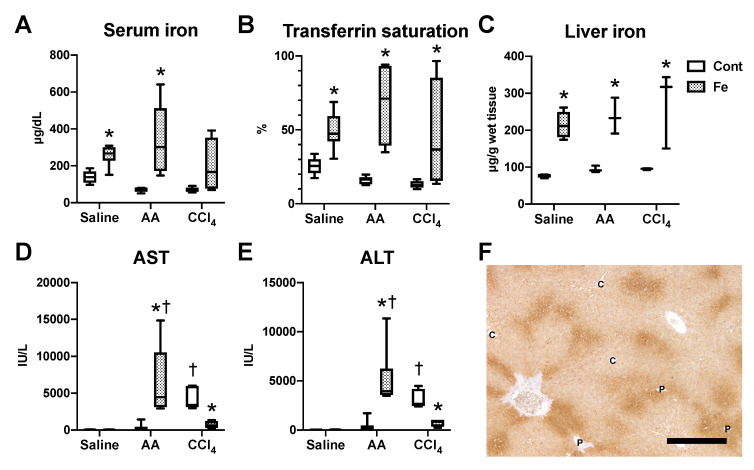
Biochemical parameters for iron metabolism (**A**–**C**) and for liver enzymes (**D**,**E**). * *p* < 0.05 vs. control diet group with same chemical administration (diet factor), † *p* < 0.05 vs. saline group with same diet feeding (chemical factor), by Sidak’s multiple comparison. Representative image of iron histochemistry in the liver of Fe-saline group (**F**). Iron deposition, stained brown with 3,3′-diaminobenzidine (DAB), is more intense in the periportal (Zone 1) hepatocytes than in the centrilobular (Zone 3) hepatocytes. C, central vein; P, portal vein. Bar = 500 μm.

**Figure 3 nutrients-12-02784-f003:**
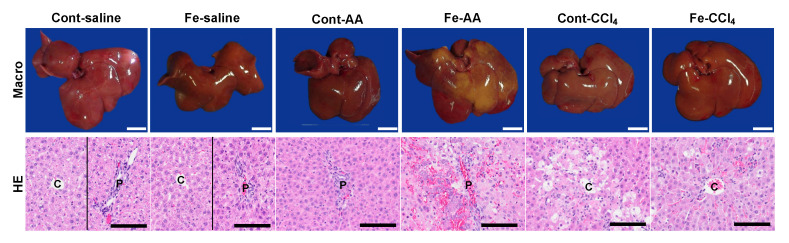
Macroscopic (upper) and hematoxylin and eosin (HE; lower) images of the liver at 24 h after administration of saline, AA, or CCl_4_. Macroscopically, extensive discoloration, corresponding extensive necrosis in HE, is seen in multiple lobules of Fe-AA group. Diffuse discoloration, consistent with centrilobular necrosis, was seen in Cont-CCl_4_ group; the microscopic lesion is less prominent in Fe-CCl_4_ group. C, centrilobular area; P, portal area. Bar = 1 cm (upper) and 100 µm (lower).

**Figure 4 nutrients-12-02784-f004:**
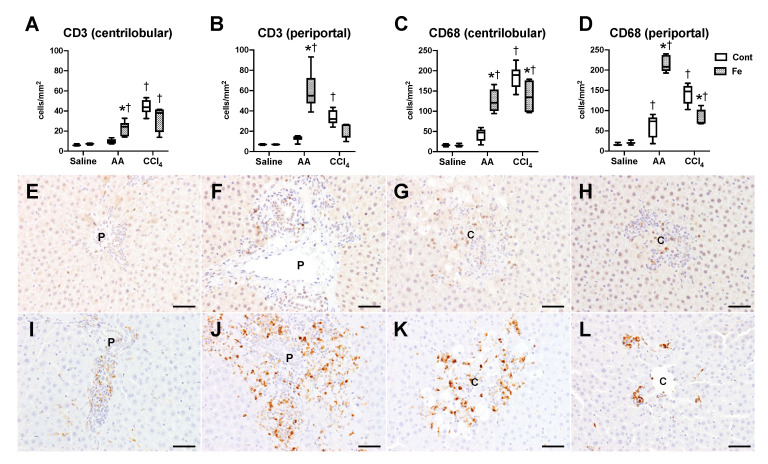
The number of CD3-positive T cells (**A**,**B**) and CD68-positive macrophages/Kupffer cells (**C**,**D**) in the centrilobular (**A**,**C**) and periportal/portal (**B**,**D**) regions. * *p* < 0.05 vs. control diet group with same chemical administration (diet factor), † *p* < 0.05 vs. saline group with same diet feeding (chemical factor), by Sidak’s multiple comparison. Representative images of immunohistochemistry for CD3 (**E**–**H**) and CD68 (**I**–**L**) in the liver of Cont-AA (**E**,**I**), Fe-AA (**F**,**J**), Cont-CCl_4_ (**G**,**K**), and Fe-CCl_4_ (**H**,**L**) groups. P, portal area; C, centrilobular area. Bar = 50 μm.

**Figure 5 nutrients-12-02784-f005:**
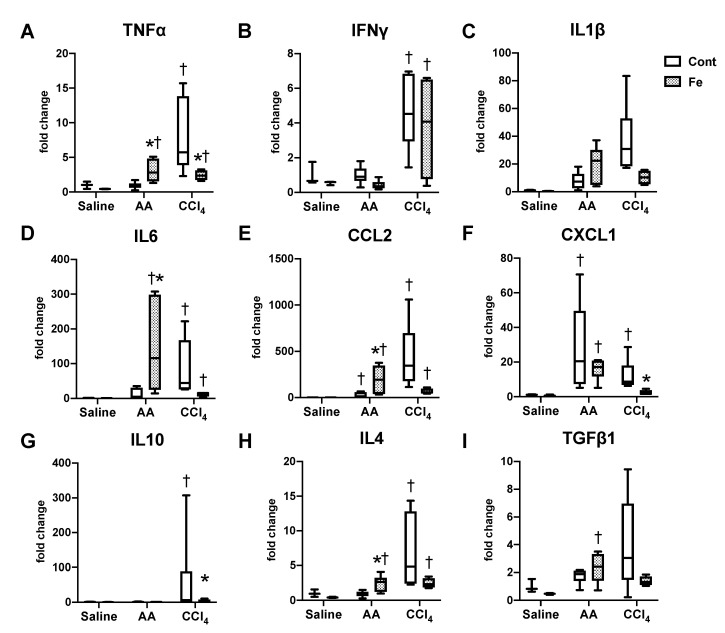
Hepatic expression of TNFα (**A**), IFNγ (**B**), IL1β (**C**), IL6 (**D**), CCL2 (**E**), CXCL1 (**F**), IL10 (**G**), IL4 (**H**), TGFβ1 (**I**). Data were normalized by the expression of 18S rRNA and are expressed as fold change to Cont-saline group. * *p* < 0.05 vs. control diet group with same chemical administration (diet factor), † *p* < 0.05 vs. saline group with same diet feeding (chemical factor), by Sidak’s multiple comparison.

**Figure 6 nutrients-12-02784-f006:**
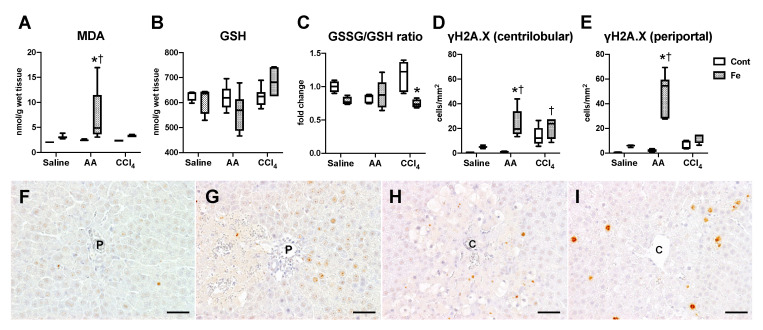
Hepatic content of malondialdehyde (MDA); (**A**), glutathione (GSH); (**B**) and the ratio of glutathione disulfide (GSSG) to GSH (**C**). The number of γH2A.X-positive hepatocytes in the centrilobular (**D**) and periportal region (**E**) of the liver. * *p* < 0.05 vs. control diet group with same chemical administration (diet factor), † *p* < 0.05 vs. saline group with same diet feeding (chemical factor), by Sidak’s multiple comparison. Representative images of immunohistochemistry for γH2A.X in the liver of Cont-AA (**F**), Fe-AA (**G**), Cont-CCl_4_ (**H**), and Fe-CCl_4_ (**I**) groups. The number of γH2A.X-positive periportal (Zone 1) hepatocytes is higher in the Fe-AA than in Cont-AA group, while the number of γH2A.X-positive centrilobular (Zone 3) hepatocytes does not differ significantly between Cont-CCl_4_ and Fe-CCl_4_ groups. P, portal vein; C, central vein. Bar = 50 μm.

**Figure 7 nutrients-12-02784-f007:**
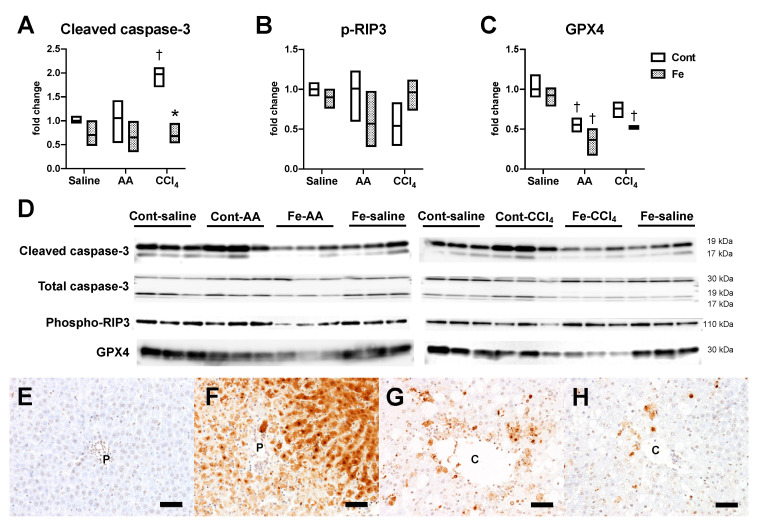
Western blot data for cleaved/total caspase-3 (**A**), phospho-RIP3 (**B**), and GPX4 (**C**). * *p* < 0.05 vs. control diet group with same chemical administration (diet factor), † *p* < 0.05 vs. saline group with same diet feeding (chemical factor), by Sidak’s multiple comparison. (**D**) shows the images of the bands analyzed. Representative images of TUNEL assay in the liver of Cont-AA (**E**), Fe-AA (**F**), Cont-CCl_4_ (**G**) and Fe-CCl_4_ (**H**) groups. P, portal vein; C, central vein. Bar = 50 μm.

**Table 1 nutrients-12-02784-t001:** The list of antibodies used for immunohistochemistry.

Antibody (Clone Name)	Type	Dilution	Pretreatment	Source	Code
CD3 (F7.2.38)	Mouse monoclonal	1:500	Microwaving in 10 mM Tris-EDTA (pH 9.0) for 20 min	Dako	M7254
CD68 (ED-1)	Mouse monoclonal	1:500	ProK (100 μg/mL) for 10 min	Millipore	MAB1435
γH2A.X (20E3)	Rabbit monoclonal	1:500	Microwaving in 10 mM citrate buffer (pH 6.0) for 20 min	Cell Signaling Technology	#9718

EDTA; ethylenediaminetetraacetic acid, ProK; proteinase K, γH2A.X; phosphorylated histone H2A.X.

**Table 2 nutrients-12-02784-t002:** The list of TaqMan probes used for real-time RT-PCR.

Symbol	Gene	Probe Assay ID	Amplicon Length (bp)
TNF-α	Tumor necrosis factor-α	Rn01525859_gl	92
CCl2	C-C motif chemokine ligand 2	Rn00580555_m1	95
IL-6	Interleukin-6	Rn01410330_ml	121
IL-4	Interleukin-4	Rn01456866_m1	128
IFN-γ	Interferon-γ	Rn00594078_m1	91
IL-10	Interleukin-10	Rn00563409_m1	70
IL-1β	Interleukin-1β	Rn00580432_ml	74
CXCL1	C-X-C motif chemokine ligand 1	Rn00578225_m1	102
TGF-β	Transforming growth factor-β	Rn00572010_ml	65
Ribosomal 18 s	Eukaryotic 18 s rRNA	Hs99999901_s1	187

**Table 3 nutrients-12-02784-t003:** The list of antibodies used for Western blot.

Antibody (Clone Name)	Type	Dilution	Source	Code
Cleaved caspase-3 (5A1E)	Rabbit monoclonal	1:1000	Cell Signaling Technology	#9664
Total caspase-3 (8G10)	Rabbit monoclonal	1:1000	Cell Signaling Technology	#9665
p-RIP3 (EPR9516(N)-25)	Rabbit monoclonal	1:1000	Abcam	ab195117
GPX4 (EPNCIR144)	Rabbit monoclonal	1:1000	Abcam	ab125066

p-RIP3, phosphorylated-receptor-interacting protein kinase 3; GPX4, glutathione peroxidase 4.

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
