# Peer review of "Dietary Iron Overload Differentially Modulates Chemically-Induced Liver Injury in Rats"

_nutrients, 2020, doi:10.3390/nu12092784_

Round 1

Reviewer 1 Report

One of the major issue with the manuscript is no-reporting of the size of the transcript which is being generated from RT-PCR. Authors are required to mention in the table the size of the transcript being generated.

Following corrections should be incorporated.

1-Line 98-99: Centrifuging blood at 4C for separation of serum from blood, causes hemolysis of RBC resulting into release of enzymes from RBC which gives erroneous result of serum enzymes levels.

2-Line 125-127: The method seems not correct “Two-point-five gram of total RNA was reverse-125 transcribed to cDNA by SuperScript VILO cDNA synthesis kit (Invitrogen, Carlsbad, CA, USA). Real-126 time PCR was performed with TaqMan gene expression assays (Life Technologies) in a PikoReal 127 Real-Time 96 PCR System (Thermo Scientific, Massachusetts, USA).”

3-Usually, RT-PCR uses micrograms of RNA, not “grams” of RNA which is mentioned in the sentence/method.

4-Line 138: Correct the “2.8. Glutathione-S-S-glutathione/glutathione-SH (GSSG/GSH) quantification” heading as under:

2.8. Reduced glutathione and oxidized glutathione (GSH/GSSG) quantification”

5-However, I don’t see that authors determined oxidized glutathione (GSSG). It should be modified accordingly.

6-Line 303-305: Convert the word “live” into “liver” in the sentence “A candidate mechanism underlying the exacerbation of AA-induced live injury in iron-overloaded liver is a combination of increased oxidative stress and increased ferroptosis.

7-Line 325-329: Change the sentences “It is known that moderate amount of oxidative stress can activate anti-oxidant molecules such as nuclear factor-erythroid 2-related factor 2 (Nrf2) [25]. Nrf2 induces transcription of antioxidant response elements (ARE), resulting in the production of numerous antioxidant enzymes, such as glutathione-s-transferase (GSTr), glutathione peroxidase (GPx), superoxide dismutase (SOD), catalase (CAT), and heme-oxygenase-1 (HO-1)”.

Into “It is known that moderate amount of oxidative stress can activate anti-oxidant response processes such as nuclear factor-erythroid 2-related factor 2 (Nrf2) [25]. Nrf2 is a transcription factor which induces transcription of antioxidant response elements (ARE) mediated transcriptional activation, resulting in the production of numerous antioxidant enzymes, such as glutathione-s-transferase (GSTr), glutathione peroxidase (GPx), superoxide dismutase (SOD), catalase (CAT), and heme-oxygenase-1 (HO-1).”

Author Response

Thank you for your valuable comment and suggestions. We have revised the manuscript according to your suggestions. The changes in the revised manuscript are highlighted in red.

Reviewer 1:

One of the major issue with the manuscript is no-reporting of the size of the transcript which is being generated from RT-PCR. Authors are required to mention in the table the size of the transcript being generated.

Response: We have added the size of the amplicon in Table 2. We also have added the reference of the study using the same Taqman probes (L136).

Following corrections should be incorporated.

1-Line 98-99: Centrifuging blood at 4C for separation of serum from blood, causes hemolysis of RBC resulting into release of enzymes from RBC which gives erroneous result of serum enzymes levels.

Response: This method is common to separate rat serum from blood. We confirmed that visible hemolysis of RBC did not occur at sampling. We have added the reference with the same serum separation in the method section (L106).

2-Line 125-127: The method seems not correct “Two-point-five gram of total RNA was reverse-125 transcribed to cDNA by SuperScript VILO cDNA synthesis kit (Invitrogen, Carlsbad, CA, USA). Real-126 time PCR was performed with TaqMan gene expression assays (Life Technologies) in a PikoReal 127 Real-Time 96 PCR System (Thermo Scientific, Massachusetts, USA).” Usually, RT-PCR uses micrograms of RNA, not “grams” of RNA which is mentioned in the sentence/method.

Response: Sorry for the mistake. We changed “gram” to “microgram.” (L133)

4-Line 138: Correct the “2.8. Glutathione-S-S-glutathione/glutathione-SH (GSSG/GSH) quantification” heading as under: “2.8. Reduced glutathione and oxidized glutathione (GSH/GSSG) quantification” However, I don’t see that authors determined oxidized glutathione (GSSG). It should be modified accordingly.

Response: We have added the data of GSSG/GSH ratio in Figure 6C. (L276-279)

6-Line 303-305: Convert the word “live” into “liver” in the sentence “A candidate mechanism underlying the exacerbation of AA-induced live injury in iron-overloaded liver is a combination of increased oxidative stress and increased ferroptosis.

Response: We changed “live” to “liver.” (L333)

7-Line 325-329: Change the sentences “It is known that moderate amount of oxidative stress can activate anti-oxidant molecules such as nuclear factor-erythroid 2-related factor 2 (Nrf2) [25]. Nrf2 induces transcription of antioxidant response elements (ARE), resulting in the production of numerous antioxidant enzymes, such as glutathione-s-transferase (GSTr), glutathione peroxidase (GPx), superoxide dismutase (SOD), catalase (CAT), and heme-oxygenase-1 (HO-1)”.

Into “It is known that moderate amount of oxidative stress can activate anti-oxidant response processes such as nuclear factor-erythroid 2-related factor 2 (Nrf2) [25]. Nrf2 is a transcription factor which induces transcription of antioxidant response elements (ARE) mediated transcriptional activation, resulting in the production of numerous antioxidant enzymes, such as glutathione-s-transferase (GSTr), glutathione peroxidase (GPx), superoxide dismutase (SOD), catalase (CAT), and heme-oxygenase-1 (HO-1).”

Response: We changed the sentences as you suggested. (L356-361)

Reviewer 2 Report

This study is original: many studies have focused on the negative effects of iron overload on liver homeostasis, but the authors focus here on the potential effects of iron deposition to prevent liver damage caused by drugs. The study was well conducted, but i have major concerns about the impact of the results. I also found methodoloy issues that need to be adressed prior to any publication. 

Major:

General:

In their experimental iron-overload model, the authors used a high-iron diet. As such, iron should deposit both in the reticuloendothelial system cells and in hepatocellular cells. In other iron-overload diseases, such as hereditary hemochromatosis, iron deposits in hepatocellular cells. In Ferroportin disease, accumulation of iron is marked in Kupffer cells, which is the hallmark of this disease. I think this is an important limitation that should be raised by the authors, because this condition may differ greatly from common iron-overload diseases in the human population (i.e. hereditary hemochromatosis and dysmetabolic iron overload syndrome).

In my opinion, authors should have used animals models of hereditary hemochromatosis and animals models of dysmetabolic iron overload syndrome, as described by Le Guenno G et al (Study of iron metabolism disturbances in an animal model of insulin resistance. Diabetes Res Clin Pract. 2007;77(3):363-370. doi:10.1016/j.diabres.2007.02.004) to keep as close as possible to DIOS, which is the most frequent iron overload disease in the population. 

Specific:

Lines 329-331: authors state that mild iron deposition may induce antioxidant enzymes. Is this hypothesis supported by any fundamental study? It sounds to me a bit counterintuitive and I am not convinced at all by this hypothesis. Authors need to improve their argumentation about this hypothesis.

Indeed, in a recent pilot study of mass-spectrometry-based lipidomic profiling, iron overload was shown to be associated with a mild inflammation (Céline Dalle et al, MS-based Lipidomic Profiling of Oxylipins Supports Mild Inflammation in Dysmetabolic Iron Overload Syndrome Affected Patients (P08-046-19), Current Developments in Nutrition, Volume 3, Issue Supplement_1, June 2019, nzz044.P08–046–19, https://doi.org/10.1093/cdn/nzz044.P08-046-19). No increase of antioxidant mediators was shown.

Moreover, if mild iron deposition had induced antioxidant enzymes, how can the authors explain the lack of scavenger properties in AA-induced oxidative stress?

Minor

Introduction section

Lines 35-36: I believe that the epidemiology of drug-induced-liver-injury may be highly variable according across the world. I suggest authors to add where this study took place.

Lines 54-55: I agree that iron uptake is tightly regulated, however I feel uncomfortable when the authors states that this regulation avoid iron deficiency. I suggest the authors to delete “deficiency” since iron deficiency is the world’s most frequent nutritional deficiency worldwide.

Lines 58-59: the formulation is ambiguous, suggesting a causal relationship between high transferrin saturation and accumulation of NTBI.

Lines 65 to 67: iron deposition may vary according to the iron-overload disease. For example, in Ferroportin disease, accumulation of iron is marked in Kupffer cells whereas in HH iron accumulation is prominent in hepatocytes. I suggest adding this in the introduction to discuss it further in the discussion section.

Methods section:

I suggest adding a figure to illustrate the research protocol with the number of rats in each group.

Results section:

Section 3.1.1: can the authors comment the lack of difference between cont and Fe fed rats in Serum iron in CCl4 group? I feel it is a simple omission because transferrin saturation and hepatic iron content appears as statistically higher in the Fe-fed rats compared to the Cont-fed rats in CCl4 group.

Section 3.1.2: did the authors compared the weight of the liver?

Author Response

Thank you for your valuable comment and suggestions. We have revised the manuscript according to your suggestions. The changes in the revised manuscript are highlighted in red.

Reviewer 2:

This study is original: many studies have focused on the negative effects of iron overload on liver homeostasis, but the authors focus here on the potential effects of iron deposition to prevent liver damage caused by drugs. The study was well conducted, but i have major concerns about the impact of the results. I also found methodoloy issues that need to be adressed prior to any publication.

Major:

General:

In their experimental iron-overload model, the authors used a high-iron diet. As such, iron should deposit both in the reticuloendothelial system cells and in hepatocellular cells. In other iron-overload diseases, such as hereditary hemochromatosis, iron deposits in hepatocellular cells. In Ferroportin disease, accumulation of iron is marked in Kupffer cells, which is the hallmark of this disease. I think this is an important limitation that should be raised by the authors, because this condition may differ greatly from common iron-overload diseases in the human population (i.e. hereditary hemochromatosis and dysmetabolic iron overload syndrome).

In my opinion, authors should have used animals models of hereditary hemochromatosis and animals models of dysmetabolic iron overload syndrome, as described by Le Guenno G et al (Study of iron metabolism disturbances in an animal model of insulin resistance. Diabetes Res Clin Pract. 2007;77(3):363-370. doi:10.1016/j.diabres.2007.02.004) to keep as close as possible to DIOS, which is the most frequent iron overload disease in the population.

Response: Thank you for the valuable suggestion. As you mentioned, the distribution of iron deposition is very important. Our main target of the present study is the condition of hepatic iron overload secondarily occurring in chronic liver diseases (e.g. NAFLD). The patterns of iron deposition in NAFLD are hepatocellular, reticuloendothelial (RES), and mixed patterns of both hepatocellular and RES; mixed pattern is most common (Hepatology. 2011;53:448-57. doi: 10.1002/hep.24038). Our iron overload model has a mixed hepatocellular and RES pattern. Of course, the animal models developing with both NAFLD and iron deposition are the best, but we believe that our model would be useful to examine how iron deposition can affect susceptibility to drug-induced hepatotoxicity in the condition of secondary iron overload. We have added sentences focusing on the pattern of iron deposition in hereditary hemochromatosis, ferroportin disease, and secondary hemochromatosis arising in CLD in the Discussion. (L316-327)

Specific:

Lines 329-331: authors state that mild iron deposition may induce antioxidant enzymes. Is this hypothesis supported by any fundamental study? It sounds to me a bit counterintuitive and I am not convinced at all by this hypothesis. Authors need to improve their argumentation about this hypothesis.

Indeed, in a recent pilot study of mass-spectrometry-based lipidomic profiling, iron overload was shown to be associated with a mild inflammation (Céline Dalle et al, MS-based Lipidomic Profiling of Oxylipins Supports Mild Inflammation in Dysmetabolic Iron Overload Syndrome Affected Patients (P08-046-19), Current Developments in Nutrition, Volume 3, Issue Supplement_1, June 2019, nzz044.P08–046–19, https://doi.org/10.1093/cdn/nzz044.P08-046-19). No increase of antioxidant mediators was shown.

Moreover, if mild iron deposition had induced antioxidant enzymes, how can the authors explain the lack of scavenger properties in AA-induced oxidative stress?

Response: We have added Western blot data on hepatic expression of heme oxygenase-1, an important anti-oxidant enzyme in the liver as Supplemental Figure 2 (L272-275). In addition, the discussion was changed to be in accordance with our substantial data. (L354-364)

Minor

Introduction section

Lines 35-36: I believe that the epidemiology of drug-induced-liver-injury may be highly variable according across the world. I suggest authors to add where this study took place.

Response: We have added information about differences of epidemiology by countries. (L35-36)

Lines 54-55: I agree that iron uptake is tightly regulated, however I feel uncomfortable when the authors states that this regulation avoid iron deficiency. I suggest the authors to delete “deficiency” since iron deficiency is the world’s most frequent nutritional deficiency worldwide.

Response: We have deleted “deficiency” as you suggested. (L56)

Lines 58-59: the formulation is ambiguous, suggesting a causal relationship between high transferrin saturation and accumulation of NTBI.

Response: We have changed the description. (L59-61)

Lines 65 to 67: iron deposition may vary according to the iron-overload disease. For example, in Ferroportin disease, accumulation of iron is marked in Kupffer cells whereas in HH iron accumulation is prominent in hepatocytes. I suggest adding this in the introduction to discuss it further in the discussion section.

Response: We added a sentence in the introduction section (L71-74) and a paragraph in the discussion section (L316-327), in accordance with your suggestion.

Methods section:

I suggest adding a figure to illustrate the research protocol with the number of rats in each group.

Response: We added the research protocol as Figure 1. (L101-103)

Results section:

Section 3.1.1: can the authors comment the lack of difference between cont and Fe fed rats in Serum iron in CCl4 group? I feel it is a simple omission because transferrin saturation and hepatic iron content appears as statistically higher in the Fe-fed rats compared to the Cont-fed rats in CCl4 group.

Response: There was no significant difference of serum iron between Cont-CCl4 group and Fe-CCl4 group (P=0.1211). However, we consider that the increased transferrin saturation and increased liver iron warrant hepatic iron overload in our model. We have added some discretion in this section. (L180-183)

Section 3.1.2: did the authors compared the weight of the liver?

Response: Unfortunately we did not measure the weight of the liver in this study. From the findings of our other studies, the weight of the liver was higher in rats with high-iron diet (0.5-1% Fe) feeding than in rats with control diet.

Reviewer 3 Report

Comments:

This paper reported iron overload differentially modulated chemical-induced liver injury, while iron exacerbated allyl alcohol-induced liver injury, it protected against carbon tetrachloride-induced liver damage. The paper is interesting but with a few small issues. 

  1. GSH is an essential antioxidant, and the ratio of GSH/GSSH determines the redox hemostasis. Thus, the authors should determine the ratio of GSH/GSSH instead of GSH only. Interestingly, in the Materials and Methods section, the authors stated the method of measuring GSSG/GSH ratio. 
  2. Please give the rat number in each group.
  3. The data are presented as Mean +/- sd, but what does the box in the plot figure mean? 
  4. P3 line 125, 2.5 grams of total RNA for cDNA preparation is too much, please confirm.
  5. P8 line 249-250, the cited figure number is wrong, should be Figure6A and 6B. 
  6. Figure 6, there are multi bands in cleaved caspase-3 and total caspase-3, which one is the right band? Please label molecular weight on the blot. Please give the loading control blot. 
  7. P5 line 171, grammar, "Serum iron increased with ... groups..." should be "...in... groups..."

Author Response

Thank you for your valuable comment and suggestions. We have revised the manuscript according to your suggestions. The changes in the revised manuscript are highlighted in red.

Reviewer 3:

This paper reported iron overload differentially modulated chemical-induced liver injury, while iron exacerbated allyl alcohol-induced liver injury, it protected against carbon tetrachloride-induced liver damage. The paper is interesting but with a few small issues.

GSH is an essential antioxidant, and the ratio of GSH/GSSH determines the redox hemostasis. Thus, the authors should determine the ratio of GSH/GSSH instead of GSH only. Interestingly, in the Materials and Methods section, the authors stated the method of measuring GSSG/GSH ratio.

Response: We have data of GSSG/GSH ratio in Figure 6C. (L276-279)

Please give the rat number in each group.

Response: We have added study design with the number of rats used in Figure 1 (requested by another reviewer). (L101-103)

The data are presented as Mean +/- sd, but what does the box in the plot figure mean?

Response: Sorry for the mistake. We changed “mean +/- SD” to “box-and-whisker plots with min to max range.” (L172)

P3 line 125, 2.5 grams of total RNA for cDNA preparation is too much, please confirm.

Response: We changed “gram” to “microgram.” (L133)

P8 line 249-250, the cited figure number is wrong, should be Figure6A and 6B.

Response: We have checked and corrected all the figure citation.

Figure 6, there are multi bands in cleaved caspase-3 and total caspase-3, which one is the right band?

Response: Procaspase-3 (30 kDa) is cleaved to large fragments (19/17 kDa) via apoptotic pathway. Thus, there are two bands in cleaved caspase-3 and three bands (pro-caspase-3 and two of its fragments) in total caspase-3. (Inayat-Hussain, S.H. et al. FEBS Lett. 1999, 456, 379-383.)

Please label molecular weight on the blot. Please give the loading control blot.

Response: We have added the molecular weight in Figure 7D. As mentioned in the method section (L164-168), total protein bands stained with Coomassie brilliant blue were used for normalization of the data, as the major loading controls (beta-actin, GAPDH, and alpha-tubulin) were all changed between the groups. The image of the total bands is shown in Supplementary Figure 1.

P5 line 171, grammar, "Serum iron increased with ... groups..." should be "...in... groups..."

Response: We have changed the sentence as you suggested. (L179)

Round 2

Reviewer 2 Report

This revised version have been significantly improved. The authors answered adequately and made numerous change.

The discussion paragraph has been particularly improved and I would like to congratulate the authors for having nuanced their hypotheses.

This version seems suitable for publication. 

Author Response

Thank you for your valuable comments and suggestions.